# Association between surgeon training grade and the risk of revision following unicompartmental knee replacement: An analysis of National Joint Registry data

Timothy J. Fowler[1]*, Nicholas R. Howells[1], Ashley W. Blom[2], Adrian Sayers[1‡], Michael R. Whitehouse[1,3‡]

1 Musculoskeletal Research Unit, Translational Health Sciences, Bristol Medical School, Southmead Hospital, Bristol, United Kingdom, 2 Faculty of Health, The University of Sheffield, Sheffield, United Kingdom, 3 National Institute for Health Research Bristol Biomedical Research Centre, University Hospitals Bristol NHS Foundation Trust, University of Bristol, Bristol, United Kingdom

‡ These authors are joint senior authors on this work.
* t.j.fowler@bristol.ac.uk

**Data Availability Statement:** The data used in the study are available from The National Joint Registry (NJR) (https://www.njrcentre.org.uk). Restrictions apply to the availability of these data, which were used under license for the current study, and are therefore not publicly available. Data access

## Abstract

### Background

Unicompartmental knee replacements (UKRs) are performed by surgeons at various stages in training with varying levels of supervision, but we do not know if this is a safe practice with comparable outcomes to consultant-performed UKR. The aim of this study was to use registry data for England and Wales to investigate the association between surgeon grade (consultant, or trainee), the senior supervision of trainees (supervised by a scrubbed consultant, or not), and the risk of revision surgery following UKR.

### Methods and findings

We conducted an observational study using prospectively collected data from the National Joint Registry for England and Wales (NJR). We included adult patients who underwent primary UKR for osteoarthritis ($n = 106,206$), recorded in the NJR between 2003 and 2019. Exposures were the grade of the operating surgeon (consultant, or trainee) and whether or not trainees were directly supervised by a consultant during the procedure (referred to as "supervised by a scrubbed consultant"). The primary outcome was all-cause revision surgery. The secondary outcome was the number of procedures revised for the following specific indications: aseptic loosening/lysis, infection, progression of osteoarthritis, unexplained pain, and instability. Flexible parametric survival models were adjusted for patient, operation, and healthcare setting factors.

We included 106,206 UKRs in 91,626 patients, of which 4,382 (4.1%) procedures were performed by a trainee. The unadjusted cumulative probability of failure at 15 years was 17.13% (95% CI [16.44, 17.85]) for consultants, 16.42% (95% CI [14.09, 19.08]) for trainees overall, 15.98% (95% CI [13.36, 19.07]) for trainees supervised by a scrubbed consultant, and 17.32% (95% CI [13.24, 22.50]) for trainees not supervised by a scrubbed consultant.

applications can be made to the NJR Research Committee. With NJR permission in place, the data underlying the presented results will be available to access via the NJR data access network. The authors of this manuscript are not the data owner and do not have permission to share the data.

**Funding:** Posts of members of the research team were funded by a contract grant from the National Joint Registry, in the form of the Lot 2 contract (FTS 010307-2022: Statistical Analysis, Support and Associated Services – MRW, AWB and AS). This study was also supported by the National Institute for Health Research (NIHR) Biomedical Research Centre at the University Hospitals Bristol NHS Foundation Trust and the University of Bristol (IS-BRC-1215-20011 – MRW and AWB). TF was supported by a NIHR Academic Clinical Fellowship. AS was supported by an MRC strategic skills fellowship (MR/L01226X/1). The funders had no role in study design, data collection and analysis, the preparation of the manuscript, or the decision to publish.

**Competing interests:** MRW, AWB and AS report holding a contract with The Healthcare Quality Improvement Partnership/The National Joint Registry in the form of the Lot 2 contract (FTS 010307-2022: Statistical Analysis, Support and Associated Services), during the conduct of the submitted work. MRW and AWB were supported by the NIHR Biomedical Research Centre at University Hospitals Bristol and Weston NHS Foundation Trust and the University of Bristol (IS-BRC-1215-20011), during the conduct of the submitted work. MRW and AWB report grants from the NIHR investigating the outcomes of joint replacement, outside the submitted work; MRW and AWB are editors of an Orthopaedic textbook for which they receive royalty payments from Taylor Francis. MRW conducts teaching on courses sponsored by Heraeus and DePuy for which his institution receives market rate payments. All other authors declare no conflicts of interest.

**Abbreviations:** BASK, British Association for Surgery of the Knee; BMI, body mass index; CCT, Certification of Completion of Training; EKS, European Knee Society; FPM, flexible parametric survival modelling; FRCS, Fellowship of the Royal College of Surgeons; IMD, index of multiple deprivation; IQR, interquartile range; KM, Kaplan–Meier; NICE, National Institute for Health and Care Excellence; NJR, National Joint Registry; NZJR, New Zealand Joint Registry; OA, osteoarthritis; PH, proportional hazard; SD, standard deviation; TKR, total knee replacement; UKR, unicompartmental knee replacement.

There was no association between surgeon grade and all-cause revision in either crude or adjusted models (adjusted HR = 1.01, 95% CI [0.90, 1.13]; $p = 0.88$). Trainees achieved comparable all-cause survival to consultants, regardless of the level of scrubbed consultant supervision (supervised: adjusted HR = 0.99, 95% CI [0.87, 1.14]; $p = 0.94$; unsupervised: adjusted HR = 1.03, 95% CI [0.87, 1.22]; $p = 0.74$).

Limitations of this study relate to its observational design and include: the potential for nonrandom allocation of cases by consultants to trainees; residual confounding; and the use of the binary variable "surgeon grade," which does not capture variations in the level of experience between trainees.

## Conclusions

This nationwide study of UKRs with over 16 years' follow up demonstrates that trainees within the current training system in England and Wales achieve comparable all-cause implant survival to consultants. These findings support the current methods by which surgeons in England and Wales are trained to perform UKR.

## Author summary

### Why was this study done?

- Unicompartmental knee replacement (UKR) is an alternative to total knee replacement (TKR) in patients with symptomatic osteoarthritis. The National Institute for Health and Care Excellence (NICE) recommends that patients with isolated medial compartment OA should be offered a choice of UKR or TKR.

- Proposed advantages of UKR over TKR include superior functional outcomes, reduced length of stay, fewer medical complications, greater cost-effectiveness, and lower mortality. However, UKR revision rates are considerably higher than TKR revision rates.

- The British Association for Surgery of the Knee (BASK) and European Knee Society (EKS) have recommended that knee surgeons should have exposure to and training in UKR.

- UKRs are performed by surgeons at different stages in training with varying levels of supervision. However, we do not know if UKRs performed by trainees last as long as those performed by fully trained consultant surgeons.

### What did the researchers do and find?

- We analysed data from the National Joint Registry for England and Wales (NJR), which is the largest joint replacement registry in the world. We included over 100,000 primary UKRs performed between 2003 and 2019.

- We were interested in whether the procedure was performed by a fully trained consultant surgeon, or a trainee. We were also interested in whether or not trainees were directly supervised by a consultant during the operation. The primary outcome was all-

cause revision surgery. We used a specialist statistical method called "flexible parametric survival modelling" to analyse the data.

- We found no association between surgeon grade and all-cause revision. Trainees achieved comparable outcomes to consultants, regardless of the level of consultant supervision.

### What do these findings mean?

- These data suggest that within the current training system in the England and Wales, UKRs performed by trainee surgeons last as long as those performed by fully trained consultant surgeons.

- The findings of this study are reassuring and support the current methods by which surgeons are trained to perform UKR in England and Wales.

- Limitations of this study relate to its observational design. A notable limitation is that we used a binary exposure (consultant, or trainee), which does not capture variations in the level of experience between trainees.

## Introduction

Unicompartmental knee replacement (UKR) is an alternative to total knee replacement (TKR) in patients with symptomatic osteoarthritis (OA) isolated to a single compartment [1]. The National Institute for Health and Care Excellence (NICE) recommends that patients in England and Wales with isolated medial compartment OA should be offered a choice of UKR or TKR [2]. The British Association for Surgery of the Knee (BASK) and European Knee Society (EKS) have recommended that knee surgeons should have exposure to and training in UKR [3]. Proposed advantages of UKR over TKR include superior functional outcomes, reduced length of stay, fewer medical complications, greater cost-effectiveness, and lower mortality [4,5]. However, UKR revision rates are considerably higher than primary TKR revision rates [6,7]. Previous studies have suggested that UKRs performed by low-volume surgeons are associated with an increased risk of revision compared to UKRs performed by experienced higher volume surgeons [8,9]. This raises the question of whether or not it is safe for these procedures to be performed by trainees.

The survival of a joint replacement, defined as the absence of revision surgery over time, is the principal metric used for comparing the longevity of implant components and is a commonly used measure of surgical performance. Our current understanding of the survival of UKRs in the context of surgical training is based on a small number of observational studies, which are discussed in our recent systematic review on this subject [10]. Bottomley and colleagues conducted an observational study of 1,084 UKRs, of which 673 (62.1%) were performed by trainees. They reported no significant difference in implant survival between the groups, with 9-year cumulative survival estimates of 93.9% and 93.0% for consultants and trainees, respectively [11]. A New Zealand Joint Registry (NZJR) study of 8,854 UKRs, of which 304 (3.4%) were performed by trainees, reported no difference in the revision rates of supervised senior trainees compared to attending surgeons [12]. This study did not report

survival estimates and the overall number of trainee cases in the cohort was insufficient to facilitate meaningful comparison between the supervised and unsupervised trainee groups. The survival of UKRs according to surgeon grade and supervision remains poorly understood. It is not clear if current training practices are safe, or whether trainees achieve comparable outcomes to consultants.

The aim of this research was to use National Joint Registry (NJR) data from England and Wales to investigate the association between surgeon grade, the supervision of trainees, and the risk of revision following UKR.

## Methods

### Patients and data sources

We performed an observational study using prospectively collected data recorded in the NJR. The initial NJR data set was 1,502,564 linked knee procedures performed between 1 April 2003 and 31 December 2019. We included primary UKRs in adult patients (aged ≥18 years) performed for an indication of OA only. Patellofemoral joint replacements were excluded. Cases were included if the operating surgeon grade was recorded as any of the following: Foundation Year 1 (F1) to Specialty Trainee Year 2 (ST2); ST3-ST8; fellow; or consultant. The process of mapping grade classifications to account for variations in terminology used in different versions of the NJR Minimum Data Set (MDS) form is outlined in S1 Appendix.

### Analysis plan

The study protocol was designed prior to commencing the study, including defining the study population, exposures, outcomes of interest, and statistical methods. The main analyses were planned prior to commencing the study and these are documented throughout this methods section. However, data-driven changes to the analysis took place and these are summarised in detail in S2 Appendix, which justifies the model selection and construction.

### Data processing

The base data set used in the current study is based on the same cut of NJR data that is used in the 17th Annual Report [13]. NJR data are annually linked to other healthcare system data sets, including Civil Registration Authority data, using unique patient identifiers. This linkage, which was carried out by the NJR prior to us obtaining the data set, is approved by the Health Research Authority under Section 251 of the NHS act 2006 [13]. The steps taken in data processing and are summarised in the study flow diagram in Fig 1 and illustrated in greater detail in S1 Fig. All exclusions are consistent with the exclusion criteria of this study and the stage at which these occurred is clearly documented.

### Exposures

The primary exposure (exposure A) was surgeon grade. This is a binary variable, which was categorised according to the grade of the operating surgeon: (1) consultant; or (2) trainee. Procedures performed by surgeons of the following grades were categorised under the variable "trainee": F1-ST2; ST3-ST8; and fellow. Consultants have completed their formal training in orthopaedic surgery and been appointed to a senior position in which they can practice independently and supervise trainees.

F1-ST2 represents the first 4 years of postgraduate training after graduating from medical school (F1, F2, ST1, and ST2). ST2 doctors who have completed the Membership of the Royal College of Surgeons (MRCS) examination are eligible to apply to Specialty Training. Specialty

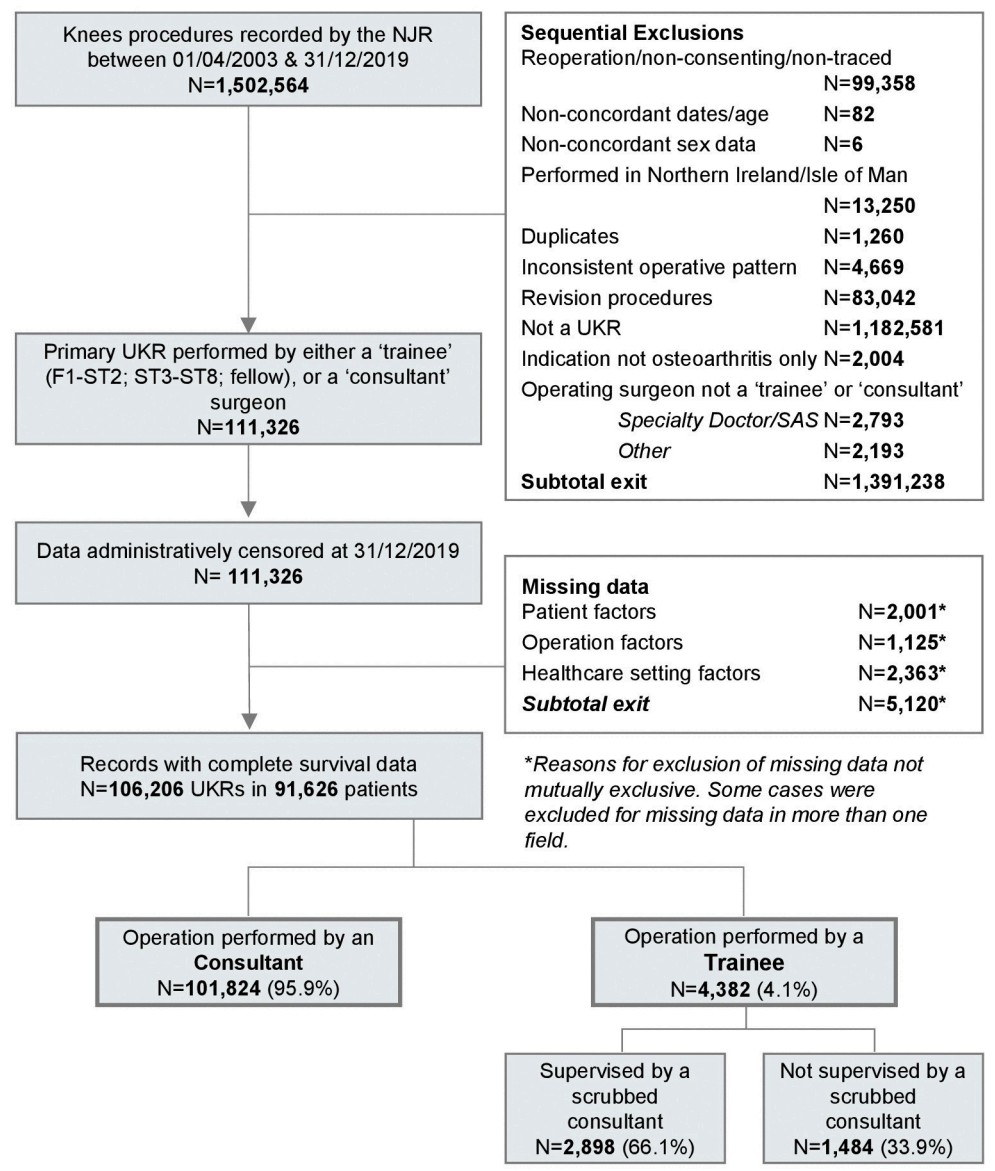

**Fig 1. Study flow diagram.**

Training in Trauma and Orthopaedic Surgery is typically a six-year programme (ST3-8). ST3-ST8 trainees are referred to as "specialty trainees," or "registrars." Progression through training levels is dependent on the successful completion of training requirements and competencies. Trainees who have completed ST6 are eligible to sit the examination for Fellowship of the Royal College of Surgeons (FRCS), which is mandatory for Certification of Completion of Training (CCT). Trainees subsequently progress to post-CCT fellowship training prior to applying for a consultant position. The term "consultant" is synonymous with "attending", and the term "registrar" is synonymous with "resident" in many healthcare settings including the United States of America. A schematic summary of the stages of surgical training in the United Kingdom is included in S3 Appendix [14].

The secondary exposure was whether or not trainees were directly supervised by a consultant during the procedure (exposure B). We refer to direct consultant supervision as

"supervised by a scrubbed consultant" throughout this paper. Therefore, trainee cases were subcategorised as follows: (1) trainee supervised by a scrubbed consultant; or (2) trainee not supervised by a scrubbed consultant. Cases were categorised as "supervised by a scrubbed consultant" if the first assistant was recorded as a consultant.

Given the variability in the level of experience between individual trainees, we performed a sensitivity analysis by recategorising cases according to the specific training grade of the operating surgeon (exposure C: consultant; F1-ST2; ST3-ST8; or fellow). Cases were further subcategorised according to the level of scrubbed consultant supervision.

## Outcomes of interest

The primary outcome was all-cause revision, which was defined as any procedure to add, remove, or modify one or more components of an implant construct for any reason [13]. The secondary outcome measure was the number of procedures revised for the following specific indications, which are listed as the 5 most common indications for knee replacement revision by the NJR: aseptic loosening/lysis, infection, progression of OA, unexplained pain, and instability [13].

## Statistical analysis

Frequencies and percentages were used to describe categorical variables. The mean, standard deviation (SD), and interquartile range (IQR) were used to describe continuous variables. Unrevised cases were either administratively censored on 31 December 2019, or the date of death, depending on which was earliest. Unadjusted estimates of net implant failure were calculated using the Kaplan–Meier (KM) method.

We performed a comprehensive exploratory analysis using Cox regression. A combination of graphical plots, Schoenfeld residuals, and likelihood ratio testing (comparing proportional and non-proportional hazards models) were used to assess the proportional hazards (PH) assumption at each level of adjustment and to assess the time-dependent effects of each confounding variable [15]. Adjusted analyses did not satisfy the PH assumption, which was due to the time-dependent effects of multiple confounding variables included in the models (age, sex, IMD, approach, fixation, bearing mobility, year of operation, and funding source). Surgeon grade (exposures A) did not demonstrate a time-dependent effect.

To account for non-proportionality, we used flexible parametric survival modelling (FPM) [15,16], which has been used in previous NJR analyses [17–20]. This method uses restricted cubic spline functions to model the baseline hazard and account for the time-dependent effects of specified variables. Graphical assessment, AIC, BIC, and likelihood ratio testing were used to optimise the fit and complexity of the final model [15]. This process of model selection, construction, and justification is described in greater detail in S2 Appendix.

Analyses were adjusted for categorical confounding variables in the following manner. Model 1 was unadjusted. Model 2 was adjusted for patient-level factors (age, sex, American Society of Anaesthesiologists (ASA) grade, and index of multiple deprivation (IMD) decile). Model 3 was further adjusted for operation-level factors (anaesthetic, approach, fixation, and bearing mobility). Model 4 was further adjusted for healthcare setting factors (funding source and year of operation). In each case, the baseline category was the most frequently occurring (as detailed in S4 Appendix).

Body mass index (BMI) is missing in a large proportion of NJR records. It has been reported that approximately 40% of patients did not have a BMI recorded in the NJR in 2009, compared to approximately 18% in 2016 [21]. Due to the significant proportion of records with missing values, BMI was not included as a confounding variable in the analyses. This is

consistent with the approach used in previous NJR studies and this decision was made prior to initiating the study based on the known pattern of missing data [22,23].

We performed separate analyses for all-cause revision and the 5 specific indications for revision (aseptic loosening/lysis, infection, progression of OA, unexplained pain, and instability), which were examined as separate survival endpoints. Separate FPM analyses were performed for each exposure and analyses were incrementally adjusted for confounding variables.

In response to the peer review process, we conducted an additional sensitivity analysis to explore the lack of independence between observations in patients who underwent bilateral procedures (on the same day, or on different days). We examined the primary outcome measure using a fully adjusted (Model 4) multilevel mixed effects parametric survival model to introduce a frailty term and account for time-dependent effects [24]. All analyses were performed using Stata (Version SE 15.1; StataCorp LP, USA). This study is reported as per the Reporting of studies Conducted using Observational Routinely collected health Data (RECORD) Statement (S1 RECORD Checklist) [25].

## Patient and public involvement

Patient representatives sit on the committee structure of the NJR. The research priorities of the NJR are identified by this committee and approved by the patient representatives. Patients were not involved in setting the research question or the outcome measures nor were they involved in the design, implementation, or interpretation of the results of this study. We are unable to disseminate the results of this study directly to study participants due to the anonymous nature of the data. We plan to disseminate our findings through the NJR communications team to relevant individuals who determine the provision of joint replacement and to the general population through local and national press.

## Ethics statement

The NJR supports public health surveillance and wider clinical decision-making and holds data that are anonymous to the researchers who use it. NHS Health Research Authority guidance dictates that the secondary use of such data for research does not require approval by a research ethics committee. Therefore, separate research ethics committee approval was not required for this study. Patients are consented for inclusion in the NJR according to standard practice, with permission under the Health Service (Control of Patient Information) Regulations, otherwise referred to as Section 251 support [26].

## Results

### Descriptive analysis

We included 106,206 UKR procedures in 91,626 patients, of which 4,382 (4.1%) were performed by trainees. Trainees were supervised by a scrubbed consultant in 66.1% ($n$ = 2,898) of trainee-performed cases (Table 1 and Fig 1).

The mean age of patients operated on by trainees was 1.7 years older than patients operated on by consultants (65.5 versus 63.8 years). Trainees operated on a lower proportion of ASA I patients (15.7% versus 21.3%) and a higher proportion of ASA ≥III patients (13.4% versus 8.4%). A higher proportion of trainee procedures utilised uncemented implants (23.6% versus 19.9%) and a mobile bearing (72.0% versus 60.9%) (Table 1).

The maximum duration of follow up was 16.8 years. Mean follow up was 6.5 years (SD 4.3; IQR 2.6 to 10.1 years) for trainee UKRs and 5.6 years (SD 4.00; IQR 2.2 to 8.6 years) for

**Table 1. Descriptive statistics for patient, operation, and healthcare setting factors for included UKRs.**

| Variable | Surgeon grade and supervision ($n = 106,206$) | | | |
|---|---|---|---|---|
| | Consultant ($n = 101,824$) | Trainee (overall) ($n = 4,382$) | Trainee supervised by a scrubbed consultant ($n = 2,898$) | Trainee not supervised by a scrubbed consultant ($n = 1,484$) |
| **Mean age (SD)** | 63.8 (9.7) | 65.5 (9.6) | 65.4 (9.7) | 65.7 (9.5) |
| **Age groups (%)** | | | | |
| <55 | 18,562 (18.2) | 594 (13.6) | 408 (14.1) | 186 (12.5) |
| 55–64 | 35,656 (35.0) | 1,416 (32.3) | 938 (32.4) | 478 (32.2) |
| 65–74 | 33,057 (32.5) | 1,541 (35.2) | 1,014 (35.0) | 527 (35.5) |
| 75–84 | 13,132 (12.9) | 745 (17.0) | 474 (16.4) | 271 (18.3) |
| >85 | 1,417 (1.4) | 86 (2.0) | 64 (2.2) | 22 (1.5) |
| **Female (%)** | 46,972 (46.1) | 2,105 (48.0) | 1,410 (48.7) | 695 (46.8) |
| **Side (%)** | | | | |
| Right | 50,989 (50.1) | 2,138 (48.8) | 1,396 (48.2) | 742 (50.0) |
| **IMD decile (%)\*** | | | | |
| 1–2 (most deprived) | 9,778 (9.6) | 496 (11.3) | 356 (12.3) | 140 (9.4) |
| 3–4 | 14,969 (14.7) | 712 (16.3) | 501 (17.3) | 211 (14.2) |
| 5–6 | 22,054 (21.7) | 886 (20.2) | 588 (20.3) | 298 (20.1) |
| 7–8 | 25,655 (25.2) | 1,013 (23.1) | 660 (22.8) | 353 (23.8) |
| 9–10 (least deprived) | 29,368 (28.8) | 1,275 (29.1) | 793 (27.4) | 482 (32.5) |
| **BMI (kg/m$^2$)** | | | | |
| <19 (underweight) | 140 (0.1) | 5 (0.1) | 3 (0.1) | 2 (0.1) |
| 19–24.9 (normal) | 8,049 (7.9) | 264 (6.0) | 190 (6.6) | 74 (5.0) |
| 25–29.9 (overweight) | 27,948 (27.5) | 1,032 (23.6) | 699 (24.1) | 333 (22.4) |
| >30 (obese) | 37,431 (36.8) | 1,571 (35.9) | 1,101 (38.0) | 470 (31.7) |
| Missing | 28,256 (27.8) | 1,510 (34.5) | 905 (31.2) | 605 (40.8) |
| **ASA grade (%)** | | | | |
| ASA I | 21,663 (21.3) | 686 (15.7) | 466 (16.1) | 220 (14.8) |
| ASA II | 71,562 (70.3) | 3,107 (70.9) | 2,021 (69.7) | 1,086 (73.2) |
| ASA ≥III | 8,599 (8.4) | 589 (13.4) | 411 (14.2) | 178 (12.0) |
| **Anaesthetic (%)** | | | | |
| Spinal | 57,928 (56.9) | 2,193 (50.1) | 1,544 (53.3) | 649 (43.7) |
| General | 47,812 (47.0) | 2,164 (49.4) | 1,380 (47.6) | 784 (52.8) |
| Epidural | 4,290 (4.2) | 331 (7.6) | 174 (6.0) | 157 (10.6) |
| Nerve block | 16,847 (16.6) | 948 (21.6) | 607 (21.0) | 341 (23.0) |
| **Approach (%)** | | | | |
| Lateral parapatellar | 3,310 (3.3) | 111 (2.5) | 89 (3.1) | 22 (1.5) |
| Medial parapatellar | 90,149 (88.5) | 3,982 (90.9) | 2,593 (89.5) | 1,389 (93.6) |
| Mid-vastus | 3,968 (3.9) | 131 (3.0) | 109 (3.8) | 22 (1.5) |
| Sub-vastus | 1,595 (1.6) | 44 (1.0) | 29 (1.0) | 15 (1.0) |
| Other | 2,802 (2.8) | 114 (2.6) | 78 (2.7) | 36 (2.4) |
| **Fixation (%)** | | | | |
| Cemented | 79,206 (77.8) | 3,208 (73.2) | 2,123 (72.2) | 1,085 (73.1) |
| Uncemented | 20,209 (19.9) | 1,036 (23.6) | 666 (23.0) | 370 (24.9) |
| Hybrid | 2,409 (2.4) | 138 (3.2) | 109 (3.8) | 29 (2.0) |
| **Bearing mobility (%)** | | | | |
| Fixed | 34,268 (33.7) | 912 (20.8) | 695 (24.0) | 217 (14.6) |
| Mobile | 62,011 (60.9) | 3,153 (72.0) | 1,978 (68.3) | 1,175 (79.2) |
| Monobloc poly tibia | 5,545 (5.5) | 317 (7.2) | 225 (7.8) | 92 (6.2) |

(*Continued*)

**Table 1.** (Continued)

| Variable | Surgeon grade and supervision ($n = 106,206$) | | | |
|---|---|---|---|---|
| | Consultant ($n = 101,824$) | Trainee (overall) ($n = 4,382$) | Trainee supervised by a scrubbed consultant ($n = 2,898$) | Trainee not supervised by a scrubbed consultant ($n = 1,484$) |
| **Funding source (%)** | | | | |
| NHS | 77,595 (76.2) | 4,370 (99.7) | 2,893 (99.8) | 1,477 (99.5) |
| Private | 24,229 (23.8) | 12 (0.3) | 5 (0.2) | 7 (0.5) |
| **Year of operation (%)** | | | | |
| 2003–2011 | 35,054 (34.4) | 2,080 (47.5) | 1,241 (42.8) | 839 (56.5) |
| 2012–2019 | 66,770 (65.6) | 2,302 (52.5) | 1,657 (57.2) | 645 (43.5) |

Data are $n$ (%) or mean (SD); denoted where applicable

*IMD deciles used for analysis.

ASA, American Society of Anaesthesiologists; BMI, body mass index; IMD, index of multiple deprivation; NHS, National Health Service; UKR, unicompartmental knee replacement.

consultant UKRs. A total of 6,920 UKRs were revised at a mean of 4.3 years (SD 3.5; IQR 1.5 to 6.5 years).

## Missing data

Details of missing data are documented in S2 Fig. Fewer than 5% of cases ($n = 5,120$) had missing data. Complete-case analysis was used in all analyses and records with missing data in any confounding variable field used in subsequent statistical models were excluded from the relevant model. This is based on the assumption that the pattern of missingness of NJR data is independent of the primary exposure and the outcome. Considering the large data sets, the small proportion of incomplete cases and the assumed pattern of missingness, any potential improvement in efficiency from using multiple imputation compared to complete-case analysis is likely to be negligible [22,27].

## All-cause revision

The unadjusted cumulative probability of failure at 15 years was 17.13% (95% CI [16.44, 17.85]) for consultants, 16.42% (95% CI [14.09, 19.08]) for trainees overall, 15.98% (95% CI [13.36, 19.07]) for trainees supervised by a scrubbed consultant, and 17.32% (95% CI [13.24, 22.50]) for trainees not supervised by a scrubbed consultant. Failure estimates (one minus survival) for all intervals of follow up are presented in Table 2, and graphically displayed as a one minus KM plot in Fig 2.

Unadjusted FPM analysis comparing UKRs performed by consultants and trainees (exposure A), indicated that surgeon grade was not associated with the risk of all-cause revision (Model 1: HR = 1.05, 95% CI [0.94, 1.17]; $p = 0.40$). This finding, which is documented in Table 3, persisted despite incremental adjustment for patient, operation, and healthcare setting factors (Model 4: HR = 1.01, 95% CI [0.90, 1.13]; $p = 0.88$). Further analysis was performed according to the level of senior supervision (exposure B). Neither crude nor adjusted models demonstrated an association between the level of supervision of trainees and the risk of all-cause revision (Table 3).

Sensitivity analysis was performed following further subcategorisation of cases according to specific training grade (exposure C) and supervision. There was no evidence of an association between any specific training grade (F1-ST2, ST3-ST8, or fellow) and an increased risk of all-

**Table 2. The unadjusted cumulative probability of all-cause failure of UKRs according to surgeon grade (exposure A) and supervision (exposure B).**

| Follow up (years) | Consultant | | | Trainee (overall) | | | Trainee supervised by a scrubbed consultant | | | Trainee not supervised by a scrubbed consultant | | |
|---|---|---|---|---|---|---|---|---|---|---|---|---|
| | Number at risk* | Number of revisions | % Failure (95% CI) | Number at risk* | Number of revisions | % Failure (95% CI) | Number at risk* | Number of revisions | % Failure (95% CI) | Number at risk* | Number of revisions | % Failure (95% CI) |
| 1 | 101,824 | 986 | 1.02 (0.96, 1.10) | 4,382 | 47 | 1.12 (0.84, 1.49) | 2,898 | 29 | 1.05 (0.73, 0.15) | 1,484 | 18 | 1.26 (0.80, 2.00) |
| 3 | 90,264 | 2,121 | 3.63 (3.51, 3.76) | 3,954 | 105 | 4.01 (3.43, 4.68) | 2,605 | 64 | 3.74 (3.06, 4.57) | 1,349 | 41 | 4.52 (3.52, 5.80) |
| 5 | 67,809 | 1,150 | 5.50 (5.34, 5.67) | 3,119 | 65 | 6.22 (5.46, 7.08) | 2,009 | 47 | 6.29 (5.34, 7.40) | 1,110 | 18 | 6.16 (4.95, 7.65) |
| 7 | 49,530 | 867 | 7.43 (7.23, 7.64) | 2,475 | 49 | 8.28 (7.36, 9.32) | 1,532 | 24 | 7.92 (6.80, 9.21) | 943 | 25 | 8.93 (7.34, 10.77) |
| 10 | 35,199 | 874 | 10.52 (10.24, 10.81) | 1,916 | 35 | 10.35 (9.23, 11.59) | 1,180 | 21 | 10.10 (8.70, 11.71) | 736 | 14 | 10.86 (9.08, 12.96) |
| 13 | 17,425 | 461 | 14.44 (13.99, 14.91) | 1,125 | 39 | 14.74 (13.05, 16.63) | 629 | 23 | 14.87 (12.62, 17.49) | 496 | 16 | 14.76 (12.30, 17.67) |
| 15 | 5,117 | 98 | 17.13 (16.44, 17.85) | 369 | 4 | 16.42 (14.09, 19.08) | 208 | 2 | 15.98 (13.36, 19.07) | 161 | 2 | 17.32 (13.24, 22.50) |

Data are the number at risk, the number of revision events, the unadjusted cumulative probability of failure and the 95% CI.

*Number at risk at the beginning of interval.

cause revision, regardless of the level of supervision (Table 4). It should be noted that very few UKRs were performed by surgeons in the most junior category (F1-ST2).

An additional sensitivity analysis was performed to explore the lack of independence between observations in patients who underwent bilateral procedures. The results were very

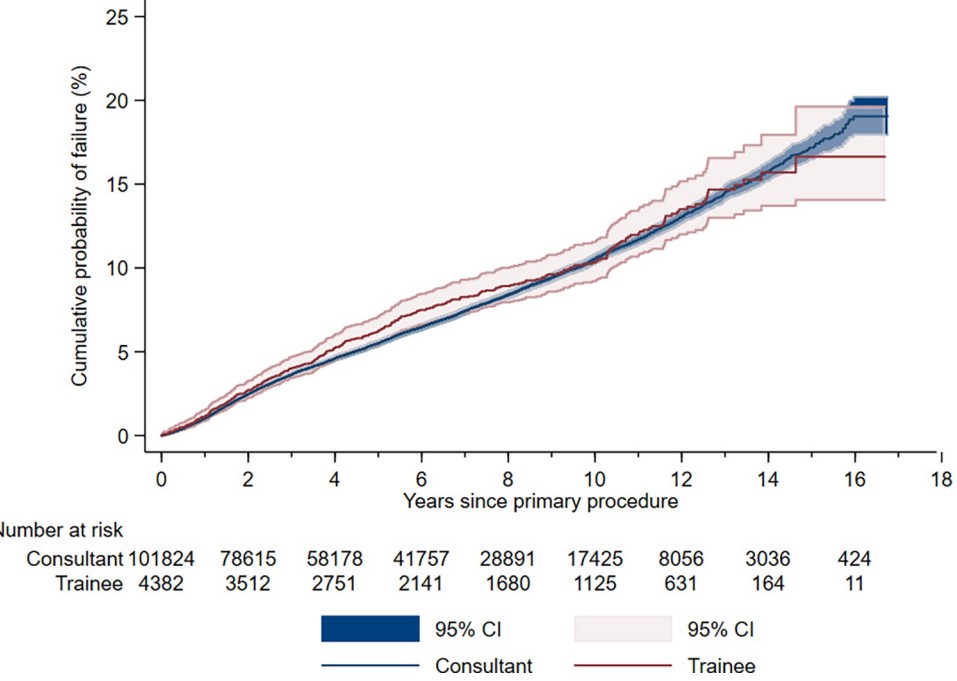

**Fig 2.** Kaplan–Meier plot (one minus survival) demonstrating the cumulative probability of UKR failure (i.e., all-cause revision) according to surgeon grade (exposure A).

**Table 3. Results of flexible parametric models (FPMs) according to surgeon grade (exposure A) and supervision (exposure B).**

| Indication for revision | Exposure subgroup | Exposure | Revisions (*n*)* | Model 1 (unadjusted) *n* = 106,206 | | | Model 2 (adjusted for †) *n* = 106,206 | | | Model 3 (adjusted for †, ‡) *n* = 106,206 | | | Model 4 (adjusted for †, ‡, §) *n* = 106,206 | | |
|---|---|---|---|---|---|---|---|---|---|---|---|---|---|---|---|
| | | | | HR | 95% CI | *p*-value | HR | 95% CI | *p*-value | HR | 95% CI | *p*-value | HR | 95% CI | *p*-value |
| **All-cause revision** | A | Consultant | 6,576 | 1.00 | | | 1.00 | | | 1.00 | | | 1.00 | | |
| | | Trainee (overall) | 344 | 1.05 | 0.94, 1.17 | 0.40 | 1.09 | 0.98, 1.21 | 0.13 | 1.05 | 0.94, 1.17 | 0.40 | 1.01 | 0.90, 1.13 | 0.88 |
| | B | Consultant | 6,576 | 1.00 | | | 1.00 | | | 1.00 | | | 1.00 | | |
| | | Trainee supervised | 210 | 1.02 | 0.89, 1.17 | 0.75 | 1.05 | 0.92, 1.21 | 0.46 | 1.03 | 0.90, 1.18 | 0.70 | 0.99 | 0.87, 1.14 | 0.94 |
| | | Trainee unsupervised | 134 | 1.09 | 0.92, 1.29 | 0.32 | 1.15 | 0.97, 1.36 | 0.12 | 1.08 | 0.91, 1.29 | 0.36 | 1.03 | 0.87, 1.22 | 0.74 |
| **Progression of OA** | A | Consultant | 2,161 | 1.00 | | | 1.00 | | | 1.00 | | | 1.00 | | |
| | | Trainee (overall) | 110 | 0.96 | 0.79, 1.16 | 0.64 | 0.98 | 0.81, 1.19 | 0.84 | 0.97 | 0.80, 1.17 | 0.77 | 0.99 | 0.82, 1.21 | 0.95 |
| | B | Consultant | 2,161 | 1.00 | | | 1.00 | | | 1.00 | | | 1.00 | | |
| | | Trainee supervised | 77 | 1.10 | 0.87, 1.37 | 0.43 | 1.11 | 0.88, 1.40 | 0.36 | 1.11 | 0.88, 1.39 | 0.38 | 1.13 | 0.90, 1.42 | 0.31 |
| | | Trainee unsupervised | 33 | 0.74 | 0.52, 1.04 | 0.08 | 0.77 | 0.54, 1.08 | 0.13 | 0.76 | 0.54, 1.07 | 0.11 | 0.78 | 0.55, 1.10 | 0.16 |
| **Aseptic loosening/ lysis** | A | Consultant | 1,877 | 1.00 | | | 1.00 | | | 1.00 | | | 1.00 | | |
| | | Trainee (overall) | 95 | 1.02 | 0.83, 1.25 | 0.86 | 1.07 | 0.87, 1.32 | 0.52 | 1.03 | 0.84, 1.27 | 0.78 | 0.96 | 0.78, 1.19 | 0.72 |
| | B | Consultant | 1,877 | 1.00 | | | 1.00 | | | 1.00 | | | 1.00 | | |
| | | Trainee supervised | 60 | 1.03 | 0.79, 1.33 | 0.84 | 1.08 | 0.83, 1.39 | 0.57 | 1.05 | 0.81, 1.36 | 0.70 | 0.99 | 0.77, 1.29 | 0.96 |
| | | Trainee unsupervised | 35 | 1.00 | 0.72, 1.41 | 0.97 | 1.06 | 0.76, 1.48 | 0.74 | 0.99 | 0.71, 1.39 | 0.97 | 0.92 | 0.65, 1.28 | 0.61 |
| **Unexplained pain** | A | Consultant | 1,236 | 1.00 | | | 1.00 | | | 1.00 | | | 1.00 | | |
| | | Trainee (overall) | 72 | 1.24 | 0.98, 1.57 | 0.08 | 1.25 | 0.98, 1.58 | 0.07 | 1.20 | 0.95, 1.53 | 0.13 | 1.08 | 0.85, 1.37 | 0.54 |
| | B | Consultant | 1,236 | 1.00 | | | 1.00 | | | 1.00 | | | 1.00 | | |
| | | Trainee supervised | 38 | 1.02 | 0.74, 1.41 | 0.90 | 1.03 | 0.75, 1.43 | 0.85 | 1.01 | 0.73, 1.39 | 0.98 | 0.92 | 0.66, 1.27 | 0.60 |
| | | Trainee unsupervised | 34 | 1.62 | 1.15, 2.28 | 0.01 | 1.63 | 1.16, 2.29 | 0.01 | 1.54 | 1.10, 2.17 | 0.01 | 1.34 | 0.95, 1.89 | 0.09 |
| **Instability** | A | Consultant | 1,052 | 1.00 | | | 1.00 | | | 1.00 | | | 1.00 | | |
| | | Trainee (overall) | 44 | 0.86 | 0.64, 1.16 | 0.33 | 0.92 | 0.68, 1.25 | 0.59 | 0.86 | 0.63, 1.16 | 0.31 | 0.80 | 0.59, 1.09 | 0.16 |
| | B | Consultant | 1,052 | 1.00 | | | 1.00 | | | 1.00 | | | 1.00 | | |
| | | Trainee supervised | 27 | 0.83 | 0.57, 1.22 | 0.35 | 0.89 | 0.61, 1.31 | 0.56 | 0.85 | 0.58, 1.24 | 0.40 | 0.80 | 0.54, 1.17 | 0.25 |
| | | Trainee unsupervised | 17 | 0.90 | 0.56, 1.46 | 0.67 | 0.97 | 0.60, 1.57 | 0.90 | 0.87 | 0.54, 1.41 | 0.57 | 0.81 | 0.50, 1.32 | 0.41 |
| **Infection** | A | Consultant | 359 | 1.00 | | | 1.00 | | | 1.00 | | | 1.00 | | |
| | | Trainee (overall) | 22 | 1.31 | 0.85, 2.02 | 0.22 | 1.32 | 0.86, 2.04 | 0.20 | 1.30 | 0.84, 2.00 | 0.24 | 1.30 | 0.84, 2.01 | 0.25 |
| | B | Consultant | 359 | 1.00 | | | 1.00 | | | 1.00 | | | 1.00 | | |
| | | Trainee supervised | 13 | 1.21 | 0.69, 2.09 | 0.51 | 1.22 | 0.70, 2.13 | 0.48 | 1.21 | 0.70, 2.11 | 0.50 | 1.22 | 0.70, 2.13 | 0.49 |
| | | Trainee unsupervised | 9 | 1.50 | 0.77, 2.90 | 0.23 | 1.51 | 0.78, 2.93 | 0.22 | 1.44 | 0.74, 2.79 | 0.29 | 1.43 | 0.73, 2.79 | 0.30 |

Data are the number of revisions for each indication, hazard ratio, 95% CI, or *p*-value.

†**Patient factors:** age; sex; ASA; IMD decile.

‡**Operation factors:** anaesthetic; approach; fixation; bearing mobility.

§**Healthcare setting factors:** funding; year of operation.

*Some cases were revised for more than one indication.

similar, and we found no evidence of an association between surgeon grade (Model 4: HR = 1.01, 95% CI [0.90, 1.13]; *p* = 0.89) and the risk of all-cause revision.

## Indication for revision

The 3 most common indications for revision in this cohort were progression of OA (*n* = 2,271), aseptic loosening/lysis (*n* = 1,972), and unexplained pain (*n* = 1,308). Crude and

**Table 4. Sensitivity analysis: Results of flexible parametric models (FPMs) for all-cause revision according to the specific training grade (exposure C) and supervision.**

| Exposure | Number of cases | Number of revisions | Complete cases (*n* = 106,206) | | |
|---|---|---|---|---|---|
| | | | HR | 95% CI | *p*-Value |
| **Model 1 (unadjusted)** | | | | | |
| Consultant | 101,824 | 6,576 | 1.00 | | |
| F1-ST2 supervised by scrubbed consultant | 25 | 2 | 2.16 | 0.70, 6.71 | 0.18 |
| F1-ST2 not supervised by scrubbed consultant | 19 | 2 | 1.17 | 0.29, 4.69 | 0.82 |
| ST3-ST8 supervised by scrubbed consultant | 2,746 | 193 | 1.04 | 0.90, 1.20 | 0.59 |
| ST3-ST8 not supervised by scrubbed consultant | 1,244 | 99 | 1.07 | 0.87, 1.30 | 0.52 |
| Fellow supervised by scrubbed consultant | 127 | 14 | 0.76 | 0.45, 1.28 | 0.31 |
| Fellow not supervised by scrubbed consultant | 221 | 33 | 1.16 | 0.83, 1.64 | 0.39 |
| **Model 2 (adjusted for †)** | | | | | |
| Consultant | 101,824 | 6,576 | 1.00 | | |
| F1-ST2 supervised by scrubbed consultant | 25 | 2 | 2.08 | 0.67, 6.45 | 0.21 |
| F1-ST2 not supervised by scrubbed consultant | 19 | 2 | 1.15 | 0.29, 4.64 | 0.84 |
| ST3-ST8 supervised by scrubbed consultant | 2,746 | 193 | 1.08 | 0.93, 1.24 | 0.32 |
| ST3-ST8 not supervised by scrubbed consultant | 1,244 | 99 | 1.12 | 0.92, 1.37 | 0.25 |
| Fellow supervised by scrubbed consultant | 127 | 14 | 0.75 | 0.45, 1.27 | 0.29 |
| Fellow not supervised by scrubbed consultant | 221 | 33 | 1.23 | 0.87, 1.73 | 0.23 |
| **Model 3 (adjusted for †, ‡)** | | | | | |
| Consultant | 101,824 | 6,576 | 1.00 | | |
| F1-ST2 supervised by scrubbed consultant | 25 | 2 | 1.99 | 0.64, 6.17 | 0.23 |
| F1-ST2 not supervised by scrubbed consultant | 19 | 2 | 1.08 | 0.27, 4.34 | 0.91 |
| ST3-ST8 supervised by scrubbed consultant | 2,746 | 193 | 1.06 | 0.91, 1.21 | 0.46 |
| ST3-ST8 not supervised by scrubbed consultant | 1,244 | 99 | 1.08 | 0.88, 1.32 | 0.45 |
| Fellow supervised by scrubbed consultant | 127 | 14 | 0.71 | 0.42, 1.19 | 0.19 |
| Fellow not supervised by scrubbed consultant | 221 | 33 | 1.11 | 0.79, 1.56 | 0.56 |
| **Model 4 (adjusted for †, ‡, §)** | | | | | |
| Consultant | 101,824 | 6,576 | 1.00 | | |
| F1-ST2 supervised by scrubbed consultant | 25 | 2 | 1.93 | 0.62, 6.01 | 0.25 |
| F1-ST2 not supervised by scrubbed consultant | 19 | 2 | 1.03 | 0.26, 4.12 | 0.97 |
| ST3-ST8 supervised by scrubbed consultant | 2,746 | 193 | 1.02 | 0.89, 1.18 | 0.74 |
| ST3-ST8 not supervised by scrubbed consultant | 1,244 | 99 | 1.03 | 0.84, 1.26 | 0.77 |
| Fellow supervised by scrubbed consultant | 127 | 14 | 0.66 | 0.39, 1.12 | 0.12 |
| Fellow not supervised by scrubbed consultant | 221 | 33 | 1.04 | 0.74, 1.46 | 0.84 |

†Patient factors: age; sex; ASA; IMD decile.

‡Operation factors: anaesthetic; approach; fixation; bearing mobility.

§Healthcare setting factors: funding; year of operation.

F1 = Foundation Year 1; ST = Specialty Trainee (number denotes year of training). F1-ST2 is the most junior category, followed by ST3-ST8.

adjusted analyses demonstrated no evidence of an association between surgeon grade (exposure A) and an increased risk of revision for any indication, including aseptic loosening/lysis, infection, progression of OA, unexplained pain, or instability (Table 3).

Further analysis was performed according to the level of trainee supervision (exposure B). We found no evidence of an increased risk of revision for any indication when trainees were supervised by a scrubbed consultant. However, both crude and adjusted analyses (Models 1–3) demonstrated that procedures performed by trainees without scrubbed consultant supervision were associated with an increased risk of revision for unexplained pain, compared to

procedures performed by consultants (Model 1: HR = 1.62, 95% CI [1.15, 2.28]; $p$ = 0.01). This was not observed in the fully adjusted model (Model 4: HR = 1.34, 95% CI [0.95, 1.89]; $p$ = 0.09) (Table 3).

## Discussion

This analysis of 106,206 primary UKRs with over 16 years' follow up represents the largest study to date of UKR outcomes in the context of surgical training. We have demonstrated that when comparing UKRs performed by consultants and trainees, there was no evidence of an association between surgeon grade and the risk of all-cause revision. Trainees achieved comparable outcomes to consultants regardless of the level of scrubbed supervision. There was no evidence that UKRs performed by trainees who were supervised by a scrubbed consultant were associated with an increased risk of revision for any specific indication (including aseptic loosening/lysis, infection, progression of OA, unexplained pain, and instability) compared to consultant-performed UKRs. We found evidence that UKRs performed by trainees who were not supervised by a scrubbed consultant were more likely to be revised for unexplained pain compared to consultant-performed UKRs. However, this was not observed in the fully adjusted model. Revision for unexplained pain following UKR has previously been attributed to low-volume surgeons, but not unsupervised trainees [8]. The most common indication for revision was progression of OA. The NJR defines revision as any procedure to add, remove, or modify one or more components of an implant construct for any reason [13]. Revision for progression of OA in the context of previous UKR implies progression of arthritis in previously unreplaced compartments of the knee. This includes procedures such as revising the UKR to a TKR, or the addition of another UKR (medial, lateral, or patellofemoral) to a previously unreplaced compartment. It does not necessarily imply failure of the individual implant components but is recorded by the NJR as a failure of the construct. We found no evidence of an association between surgeon grade, or the level of supervision and the risk of revision for progression of OA, which suggests that trainers are selecting appropriate cases for their trainees.

We included over 100,000 UKRs, which makes this significantly larger than any previous study of the association between surgeon grade and UKR outcomes [10,28]. Despite limiting our study period to predate the anomalous period of elective orthopaedic practice during the COVID-19 pandemic, our findings are current and represent UKRs with over 16 years of follow up. The data were recorded in a mandatory, nationwide prospective register, which improves the external validity and generalisability of our findings by reducing sampling bias. We employed FPM to model the time-dependent effects of confounding variables and account for non-proportionality. Furthermore, our incremental approach to confounding adjustment increases transparency by demonstrating the relative contribution of patient, operation, and healthcare setting factors to the adjusted results.

Despite these strengths, our study has limitations. This is an observational study and there is likely to be a nonrandom allocation of cases by consultants to trainees. We have attempted to account for this by adjusting for a comprehensive range of confounding variables. However, we acknowledge that there may be residual confounding and confirm that, to our knowledge, there are no further steps to take to adjust for factors that might have influenced the allocation of cases. While this may make it difficult to understand what the true training effect is, our results suggest that the current process of allocating UKR cases to trainees in England and Wales is safe and effective. Implant survival is an important objective metric of success. However, we did not consider other measures that may be relevant when evaluating the success of a joint replacement, such as patient-reported outcome measures, or postoperative complications other than failure, as they are not currently reported by the NJR. OA was the only indication,

which along with adjustment for confounding variables, accounts for measurable variations in case complexity between the groups. However, our findings remain susceptible to residual confounding. For example, we did not adjust for BMI which, consistent with other NJR studies, was missing in a high proportion of records [21]. We performed multiple testing for various reasons for revision which may account for the association between unsupervised trainees and revision for unexplained pain that attenuated with adjustment. The distinction between medial and lateral UKRs is not routinely reported by the NJR. The NJR data collection process did not distinguish between medial and lateral UKRs until the introduction of MDS version 7 in 2018 and this information was not available within the data set [13].

The binary variable "surgeon grade" does not capture variations in the level of experience between individual trainees. We have attempted to address this through sensitivity analysis, by categorising cases according to the specific training grade of the surgeon; however, this categorical variable has similar limitations. Furthermore, supervision is recorded by the NJR as a binary variable according to the grade of the first assistant, which does not capture the spectrum of supervision that is necessary in the training process [29]. Thus, these categorical variables do not account for procedures that may have been part-performed by a trainee, or in which a trainee was supervised by an unscrubbed consultant.

A recent systematic review identified a small number of observational studies relating to this subject [10]. In their NZJR study, Storey and colleagues found no significant difference in the revision rate of UKRs performed by supervised senior trainees ($n = 276$) compared to attending surgeons ($n = 8,550$). They also reported that supervised senior trainees achieved comparable functional outcomes (Oxford Knee Score) to attending surgeons at 6 months. With only 14 cases in each group, the authors acknowledge that they had insufficient data for any meaningful analysis of the outcomes of UKRs performed by supervised junior trainees and unsupervised senior trainees. Furthermore, the indication for revision was not reported, and the description of the statistical methodology employed is limited [12]. Of note, a similarly low proportion of UKRs are recorded as performed by trainees in the NZJR (3.3%) and NJR (4.1%).

Bottomley and colleagues conducted a single-centre observational study of 1,084 Oxford medial UKRs (Zimmer Biomet, Swindon, UK). Trainees performed 673 UKRs (62.1%) and were supervised by a scrubbed consultant in 48% of cases. They reported no difference in implant survival between the groups, with 9-year cumulative survival estimates of 93.9% (95% CI [90.2, 97.6]) and 93.0% (95% CI [90.3, 95.7]) for consultants and trainees, respectively. In a subgroup analysis, they showed that trainees who had performed fewer than 10 UKRs had a failure rate of 5.1% compared to a failure rate of 4.7% in those who had undertaken more than 10 UKRs; a difference that was not statistically significant [11].

In comparison to the existing literature, the current study is significantly larger, has methodological advantages, longer follow up, and provides novel insight into the importance of scrubbed consultant supervision. Our findings are generally concordant with published data from another national joint registry [12], which suggests that our findings might be generalisable to other countries.

Our findings suggest that current training practices for UKR in England and Wales are safe, when defined by equivalence of survival outcomes. However, only a small proportion of UKRs in these countries are performed by trainees and it should be noted that very few UKRs were performed by surgeons of the most junior specific training grade (F1-ST2). It is likely that UKRs are typically performed by more experienced, senior trainees. However, we were unable to quantify this in the current study, due to the broad categories used by the NJR to record the grade of the operating surgeon.

It is presumed that trainers select appropriate cases for their trainees and permit trainees to operate without scrubbed supervision only when they have reached a subjective threshold of expertise commensurate with safe independent surgical practice. Our study suggests that in this context, trainees achieve comparable all-cause UKR survival to consultant surgeons. In terms of revision for unexplained pain, trainees might achieve their best outcomes when supervised by a scrubbed consultant. However, this association was not observed in the fully adjusted analysis. We propose that trainees should ideally be supervised by a scrubbed consultant when performing UKR, particularly during the early stages of training. When experienced senior trainees operate without scrubbed supervision, careful case selection is required, and scrubbed consultant supervision should be readily available.

The findings of this study are reassuring and support the current methods by which surgeons are trained to perform UKR in England and Wales. This is of particular importance in the context of current NICE guidelines, which recommend that patients with isolated medial compartment OA should be offered a choice of UKR or TKR [2]. This requires future generations of surgeons to be trained in both procedures, or for there to be easily accessible referral networks in place to allow surgeons that do not perform UKR to refer appropriate patients on to surgeons that do.

## Conclusion

This nationwide study of UKRs with over 16 years' follow up demonstrates that trainees in England and Wales achieve comparable all-cause implant survival to consultants. Our findings support the current methods by which surgeons in England and Wales are trained to perform UKR.

## Supporting information

**S1 RECORD Checklist. Reporting of studies Conducted using Observational Routinely collected health Data (RECORD) Checklist.**
(DOCX)

**S1 Fig. Detailed study flow diagram showing sequential exclusions.**
(TIF)

**S2 Fig. Detailed study flow diagram showing exclusion of missing data.**
(TIF)

**S1 Appendix. Process of accounting for changes in NJR operating surgeon grade categories.**
(DOCX)

**S2 Appendix. Model selection, construction and justification.**
(DOCX)

**S3 Appendix. Schematic summary of surgical training in the UK.**
(DOCX)

**S4 Appendix. Model specification summarising the exposures and confounding variables used in the analyses.**
(DOCX)

## Acknowledgments

We thank the patients and staff of all the hospitals who have contributed data to the National Joint Registry. We are grateful to the Healthcare Quality Improvement Partnership (HQIP), the National Joint Registry Steering Committee (NJRSC), and staff at the NJR Centre for facilitating this work.

The views expressed in this publication are those of the authors and do not necessarily reflect those of the NHS, the NIHR, the UK Department of Health and Social Care, the NJRSC, or the HQIP.

## Author Contributions

**Conceptualization:** Timothy J. Fowler, Ashley W. Blom, Adrian Sayers, Michael R. Whitehouse.

**Data curation:** Timothy J. Fowler, Adrian Sayers.

**Formal analysis:** Timothy J. Fowler, Adrian Sayers.

**Funding acquisition:** Timothy J. Fowler, Ashley W. Blom, Michael R. Whitehouse.

**Investigation:** Timothy J. Fowler, Adrian Sayers, Michael R. Whitehouse.

**Methodology:** Timothy J. Fowler, Ashley W. Blom, Adrian Sayers, Michael R. Whitehouse.

**Project administration:** Timothy J. Fowler, Michael R. Whitehouse.

**Resources:** Timothy J. Fowler, Ashley W. Blom, Adrian Sayers, Michael R. Whitehouse.

**Software:** Timothy J. Fowler, Adrian Sayers.

**Supervision:** Ashley W. Blom, Adrian Sayers, Michael R. Whitehouse.

**Validation:** Timothy J. Fowler, Michael R. Whitehouse.

**Visualization:** Timothy J. Fowler, Ashley W. Blom, Adrian Sayers, Michael R. Whitehouse.

**Writing – original draft:** Timothy J. Fowler.

**Writing – review & editing:** Timothy J. Fowler, Nicholas R. Howells, Ashley W. Blom, Adrian Sayers, Michael R. Whitehouse.

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
