## [Editor Report · Decision Letter 0]

4 Mar 2024

Dear Dr Fowler, 

Thank you for submitting your manuscript entitled "Association between surgeon training grade and risk of revision following unicompartmental knee replacement: an analysis of a National Joint Registry Data" for consideration by PLOS Medicine.

Your manuscript has now been evaluated by the PLOS Medicine editorial staff as well as by an academic editor with relevant expertise and I am writing to let you know that we would like to send your submission out for external peer review.

Please re-submit your manuscript within two working days, i.e. by Mar 06 2024 11:59PM.

Kind regards,

Syba Sunny MBBS MRes FRCPath

Associate Editor

PLOS Medicine

ssunny@plos.org

---

## [Decision Letter · Decision Letter 1]

12 Apr 2024

Dear Mr Fowler,

Many thanks for submitting your manuscript “Association between surgeon training grade and the risk of revision following unicompartmental knee replacement: an analysis of a National Joint Registry Data” (PMEDICINE-D-24-00668R1) to PLOS Medicine. The paper has been reviewed by three subject experts and a statistician; their comments are included below and can also be accessed here:

[LINK]

As you will see, the reviewers were positive about the paper but they raised a number of questions about specific study details and the methodological approach. After discussing the paper with the editorial team and an academic editor with relevant expertise, I’m pleased to invite you to revise the paper in response to the reviewers’ comments. We plan to send the revised paper to some of all of the original reviewers*, and of course we cannot provide any guarantees at this stage regarding publication.

When you upload your revision, please include a point-by-point response that addresses all of the reviewer and editorial points, indicating the changes made in the manuscript and either an excerpt of the revised text or the location (e.g.: page and line number) where each change can be found (please include the general editorial points in your response). Please submit a clean version of the paper as the main article file and a version with changes marked should as a marked-up manuscript. Please also check the guidelines for revised papers at http://journals.plos.org/plosmedicine/s/revising-your-manuscript for any that apply to your paper.

We expect to receive your revised manuscript by May 3rd. If this deadline does not seem feasible, please let me know and we can discuss a suitable alternative.

Please don’t hesitate to contact me directly with any questions (ssunny@plos.org). 

Kind regards,

Syba

Syba Sunny MBBS, MRes, FRCPath

Associate Editor

PLOS Medicine

plosmedicine.org

ssunny@plos.org

Editorial comments:

The editorial team all agree that you present a very interesting study and, we are grateful that you gave us the opportunity to consider your work. We are pleased that the peer reviews were positive. However, we do also agree with the reviewers’ points regarding your data analyses (please see below) which could be more robust and additionally informative. All reviewer comments will need to be addressed before we can consider the manuscript further. Please see below for specific comments and respond in full.

Comments from the reviewers:

Reviewer #1: 

This study uses a national joint registry to compare the revision rate of UKR when the primary surgeon was either a trainee or a consultant to establish whether the current training model is appropriate with respect to the longevity of the implant. The conclusions were that there is little difference between the revision outcomes between trainees and consultants, including those cases where the trainee is "supervised" or "non supervised" and that the current training model does not disadvantage patients.

Generally the methodology is satisfactory within the framework of using registry data. The authors do not distinguish between lateral and medial UKR which I find a little unusual when we know that lat UKR especially with a mobile bearing are technically more difficult with a higher revision rate. It would be interesting to know if there is indeed a difference in the UK. If the authors cannot easily separate the med from lat UKRs then I would like to see a section relating to this in the limitation section.

Progression of OA was the commonest cause for revision and I would like the authors to expand on the implications of this cause for revision within the framework of the study - progression of OA is hardly a failure of the implant and more likely to be due to inappropriate patient selection (overstuffing may have occurred but is probably much less likely to cause progression of OA especially as we know that there continues to be subsidence of both cemented and uncemented implants over time), not being aware of OA elsewhere at the time of surgery, which is likely to be a decision event which is outside of the trainee's brief. Although the actual revision rate for progression of OA is similar across all surgeons a comment on this would be useful.

The authors comment on the limitation of no PROMs data which is a significant limitation, especially when considering unexplained pain, however within the confines of registry data this is acceptable.

Otherwise I believe the authors should be congratulated for a robust and sound study.

Reviewer #2: See attachment

Michael Dewey

Reviewer #3: 

The authors evaluated patient outcomes following unicompartmental knee joint replacement surgery upon the variable of whether the surgeon was a consultant or trainee and demonstrated that there was no difference in the primary outcome - need for revision surgery with long term follow up. This is a national registry study and is subject to limitations which the authors transparently discussed. Given that this is the most common method of training surgical trainees, I think this is a valuable contribution to the literature and provides insight into potential risk or lack thereof with our current methods. 

Reviewer #4: 

The authors investigated the revision rate of UKA in the NJR, by controlling for trainee vs consultants. 

Please add line numbers to facilitate review. 

The main issue is controlling for consultant surgeon, in light of the data on UKA.

There is sufficient evidence demonstrating that UKA should be performed on a regular basis to decrease revision. Thus, comparing low volume consultant surgeons with trainees might omit the true data. 

In these papers, the cut-off of consultant with at least 100 knee arthroplasty was used, which is a more fair comparison. I suggest adding this analysis.

10.1007/s00167-021-06650-4

The paper is otherwise well written.

[LINK]

1. Please upload any figures associated with your paper as individual TIF or EPS files with 300dpi resolution at resubmission; please read our figure guidelines for more information on our requirements: http://journals.plos.org/plosmedicine/s/figures. While revising your submission, please upload your figure files to the PACE digital diagnostic tool, https://pacev2.apexcovantage.com/. PACE helps ensure that figures meet PLOS requirements. To use PACE, you must first register as a user. Then, login and navigate to the UPLOAD tab, where you will find detailed instructions on how to use the tool. If you encounter any issues or have any questions when using PACE, please email us at PLOSMedicine@plos.org.

4. Data Availability - thank you for including a statement regarding data availability. As the data are not freely available, please include an appropriate contact (web or email address) for inquiries to the National Joint Registry. Please note that this cannot be a study author.

5. Please ensure that the study is reported according to the RECORD (STROBE if you feel it is more appropriate) guideline and include the completed RECORD checklist as Supporting Information. Please add the following statement, or similar, to the Methods: "This study is reported as per the Reporting of studies Conducted using Observational Routinely-collected health Data (RECORD) Statement (S1 Checklist)."

6. Statistical reporting

Throughout, including the abstract, please quantify the main results with 95% CIs and p values.

When reporting p values, please report as <0.001 and where higher as p=0.002, for example. 

When reporting 95% CIs, please separate upper and lower bounds with commas instead of hyphens as the latter can be confused with reporting of negative values.

Please include the actual amounts and/or absolute risk(s) of relevant outcomes (including NNT or NNH where appropriate), not just relative risks or correlation coefficients. (example for absolute risks: PMID: 28399126).

Please include any important dependent variables that are adjusted for in the analyses.

7. We appreciate that this is a retrospective study, but did the study have a prospective protocol or analysis plan? Please state this (either way) early in the Methods section. 

For all observational studies, in the manuscript text, please indicate: (1) the specific hypotheses you intended to test, (2) the analytical methods by which you planned to test them, (3) the analyses you actually performed, and (4) when reported analyses differ from those that were planned, transparent explanations for differences that affect the reliability of the study's results. If a reported analysis was performed based on an interesting but unanticipated pattern in the data, please be clear that the analysis was data-driven.

8. Abstract - please structure your abstract using the PLOS Medicine headings (Background, Methods and Findings, Conclusions).

9. Author summary - at this stage, we ask that you include a short, non-technical Author Summary of your research to make findings accessible to a wide audience that includes both scientists and non-scientists. The authors summary should consist of 2-3 succinct bullet points under each of the following headings:

• Why Was This Study Done? Authors should reflect on what was known about the topic before the research was published and why the research was needed.

• What Did the Researchers Do and Find? Authors should briefly describe the study design that was used and the study’s major findings. Do include the headline numbers from the study, such as the sample size and key findings. 

• What Do These Findings Mean? Authors should reflect on the new knowledge generated by the research and the implications for practice, research, policy, or public health. Authors should also consider how the interpretation of the study’s findings may be affected by the study limitations. In the final bullet point of ‘What Do These Findings Mean?’, please describe the main limitations of the study in non-technical language.

The Author Summary should immediately follow the Abstract in your revised manuscript. This text is subject to editorial change and should be distinct from the scientific abstract. Please see our author guidelines for more information: https://journals.plos.org/plosmedicine/s/revising-your-manuscript#loc-author-summary

10. Introduction - please ensure that you address past research and explain the need for and potential importance of your study. Indicate whether your study is novel and how you determined that. If there has been a systematic review of the evidence related to your study (or you have conducted one), please refer to and reference that review and indicate whether it supports the need for your study.

11. Discussion – please ensure that you present and organize the Discussion as follows: a short, clear summary of the article's findings; what the study adds to existing research and where and why the results may differ from previous research; strengths and limitations of the study; implications and next steps for research, clinical practice, and/or public policy; one-paragraph conclusion. Please avoid the use of sub-headings such that the discussion reads as continuous prose.

To submit your revised manuscript please use the following link:

---

## [Decision Letter · Decision Letter 2]

5 Jun 2024

Dear Mr. Fowler,

Thank you very much for re-submitting your manuscript "Association between surgeon training grade and the risk of revision following unicompartmental knee replacement: an analysis of a National Joint Registry Data" (PMEDICINE-D-24-00668R2) for review by PLOS Medicine.

We are grateful for your detailed responses to the editorial and reviewer comments. I am pleased to say that the reviewers were satisfied with your revision. Please see below for further comments which we require that you address prior to publication.

We expect to receive your revised manuscript within 2 weeks. Please email us (ssunny@plos.org or plosmedicine@plos.org) if you have any questions or concerns.

Please note, at the point of acceptance, an uncorrected proof of your manuscript will be published online ahead of the final version, unless you've already opted out via the online submission form. If, for any reason, you do not want an earlier version of your manuscript published online or are unsure if you have already indicated as such, please let the journal staff know immediately at plosmedicine@plos.org.

We look forward to receiving the revised manuscript by Wednesday 19th June.   

Sincerely,

Syba

Syba Sunny, MBBS, MRes, FRCPath

Associate Editor 

PLOS Medicine

ssunny@plos.org

Comments from Reviewers:

Reviewer #1: I am satisfied that the authors have addressed my concerns satisfactorily

Reviewer #2: The authors have addressed all my points

Reviewer #4: The authors performed a thorough revision.

Editorial comments:

GENERAL

Many of the editorial requests detailed below pertain to specific formatting and content requirements. Some may have already been incorporated into the manuscript and some may not apply, but please review the complete list of items and ensure that each item is included as necessary.

Our Academic Editor commented that your rebuttal letter was ‘thoughtful and thorough’ and only had one small suggestion at this stage: that your abstract should report absolute risks and not just relative risks. 

OBSERVATIONAL STUDIES

In the manuscript text, please indicate: (1) the specific hypotheses you intended to test, (2) the analytical methods by which you planned to test them, (3) the analyses you actually performed, and (4) when reported analyses differ from those that were planned, transparent explanations for differences that affect the reliability of the study's results. If a reported analysis was performed based on an interesting but unanticipated pattern in the data, please be clear that the analysis was data-driven.

DATA AVAILABILITY

Please provide a URL or email address for data applications to the Census Bureau.

COMPETING INTERESTS

Thank you for acknowledging that members of the research team were funded by a contract grant from the National Joint Registry (NJR). Could you provide more details please? What does this contract entail? The funding statement should include: specific grant numbers, initials of authors who received each award, URLs to sponsors’ websites, etc. I see that you write that the NIHR had no role in the design and conduct of the study, etc – I would be grateful if you could include a similar statement (where it applies) with regards to the NJR.

ABSTRACT

In the final sentence of the abstract methods and findings section, please detail the limitations of the study.

AUTHOR SUMMARY

Thank you for including this Author Summary.

Line 7 – Please briefly expand on the statement that ‘there is a growing demand for surgeons to be trained in [UKR]’ – it would be useful for readers to know why there is such a demand within this summary.

Line 11 onwards – Please re-phrase your sentences here to help readers who do not have specialist knowledge to better understand exactly what you did and/or why. For example, ‘parametric survival models’ or ‘scrubbed consultants’ may not be well-understood. Please revise for accessibility to the non-scientific reader avoiding the use of what might be considered medical and/or scientific ‘jargon’.

In the final bullet point of ‘What Do These Findings Mean?’, please describe the main limitations of the study in non-technical language.

Line 21 – suggest ‘These data suggest…’ or similar.

INTRODUCTION

Our journal readership is global – please consider revising the sentences here to reflect that you are referring to UK-based observations and guidelines.

STATISTICAL REPORTING

Throughout, please quantify the main results with 95% CIs and p values.

When reporting p values please report as <0.001 and where higher as p=0.002, for example. When reporting 95% CIs please separate upper and lower bounds with commas instead of hyphens as the latter can be confused with reporting of negative values.

Please include the actual amounts and/or absolute risk(s) of relevant outcomes (including NNT or NNH where appropriate), not just relative risks or correlation coefficients. (Example for absolute risks: PMID: 28399126).

DISCUSSION

Please re-phrase ‘We included over 100,000 knees…’ to ‘We included over 100,000 knee operations…’ (or similar).

TABLES and FIGURES

Throughout, including the supporting files, please provide titles/captions/footnotes which clearly describe the table/figure content without the need to refer to the text.

Please ensure all abbreviations including those used for statistical reporting are also clearly defined in the footnote.

Throughout please indicate whether your analyses are adjusted or unadjusted and where adjusted analyses are presented please also present unadjusted analyses for comparison. 

Please also ensure to clearly detail in the footnote/caption the factors which are adjusted for.

Please refer to https://journals.plos.org/plosmedicine/s/figures#loc-pages for further guidance.

Please consider avoiding the use of green and/or red to make your figures more accessible to those with colour blindness.

REFERENCES

In the bibliography please ensure that you list up to but no more than 6 author names followed by et al.

For all web references please ensure you include an, ‘Accessed [date].’

Journal name abbreviations should be those listed in the National Center for Biotechnology Information (NCBI) databases.

For further guidance, please see https://journals.plos.org/plosmedicine/s/submission-guidelines#loc-references

MAIN TEXT - MISCELLANEOUS

Throughout your manuscript, please avoid the use of the term ‘retrospective’ to describe your study and instead refer to it as ‘observational’. In the Abstract, for example. Please check and amend throughout.

Line 245, 261, 271 – Please remove the data availability, funding and conflict of interest statement from the main manuscript and include these only in the manuscript submission form when you re-submit.

Line 280 – please move the ethics statement to the methods section of the main manuscript.

SUPPORTING INFORMATION

Please ensure you apply all guidance detailed above to the supporting information files.

Please cite your Supporting Information as outlined here: https://journals.plos.org/plosmedicine/s/supporting-information

In the published article, supporting information files are accessed only through a hyperlink attached to the captions. For this reason, you must list captions at the end of your manuscript file. You may include a caption within the supporting information file itself, as long as that caption is also provided in the manuscript file. Do not submit a separate caption file.

Please ensure that all guidance above is applied as relevant to the supporting files.

SOCIAL MEDIA

To help us extend the reach of your research, if not already done so, please detail any X (formerly Twitter) handles you wish to be included when we tweet this paper (including your own, your coauthors’, your institution, funder, or lab) in the manuscript submission form when you re-submit the manuscript.

---

## [Editor Report · Decision Letter 3]

10 Jul 2024

Dear Mr Fowler,

Thank you very much for re-submitting your manuscript "Association between surgeon training grade and the risk of revision following unicompartmental knee replacement: an analysis of a National Joint Registry Data" (PMEDICINE-D-24-00668R3) for review by PLOS Medicine.

I have discussed the paper with my colleagues and the academic editor. I am pleased to say that provided the remaining editorial and production issues are dealt with, we are planning to accept the paper for publication in the journal.

The remaining issues that need to be addressed are listed at the end of this email. 

In revising the manuscript for further consideration here, please ensure you address the specific points made by the editors. In your rebuttal letter you should indicate your response to the reviewers' and editors' comments and the changes you have made in the manuscript. Please submit a clean version of the paper as the main article file. A version with changes marked must also be uploaded as a marked up manuscript file.

We expect to receive your revised manuscript within 1 week. Please email me directly if you have any questions or concerns.

We look forward to receiving the revised manuscript by Jul 17 2024 11:59PM.   

Sincerely,

Syba

Dr Syba Sunny, MBBS, MRes, FRCPath

Associate Editor 

PLOS Medicine

ssunny@plos.org

Requests from Editors:

Many thanks again for submitting your revised manuscript. I have discussed your response to ‘Comment 1’ with the academic editor and he would be happy for you to report the cumulative probabilities in your abstract. In addition to this, I have detailed below some other points that I would ask you to address.

Title: Please revise the latter part of your title for clarity, e.g. replace ‘an analysis of a National Joint Registry Data’ with ‘an analysis of National Joint Registry Data’ or similar.

Please mention the location (i.e. England and Wales) in the Background section of the Abstract. 

We suggest that you revise the statement in your abstract that reads ‘Unsupervised trainee cases were associated with an increased risk of revision for unexplained pain compared to consultant-performed UKRs, in all but the fully adjusted model’. Given that there was no difference in the fully adjusted model, it would seem to us that essentially there was no real difference to be found here. 

Please replace ‘all cause’ with ‘all-cause’ (i.e. with a hyphen) throughout your manuscript.

Thank you for revising the phrase ‘scrubbed consultant’ in line 11 of your earlier version of the manuscript. Please revise the phrase in the Methods and Findings section of your abstract too.

In the conclusions section of your abstract, please insert an apostrophe after the word ‘years’ in the phrase ‘16 years follow up’, so it reads ‘16 years’ follow up’.

In your Author Summary, could you briefly expand here why UKR is better than the more usual alternative; why is this recommended by NICE, etc?

Under your Exposures subheading, you mention ‘scrubbed consultants’. This is completely acceptable here, but it might make better reading if you qualify what this term means in the first instance of mentioning it and then state something along the lines of ‘henceforth referred to as scrubbed consultants…’ (or similar).

On page 11 of your manuscript, replace the square brackets for ASA, IMD and BMI with parentheses, i.e. ().

On page 12, line 6, there is a missing word ‘in’. The sentence ‘…more were they involved the design’ should have an ‘in’ before the word ‘the’.

In the section named Contributors, I note that the initials used for the authors used is not consistent with those used in the Financial Disclosures section – could this be revised for consistency please?

---

## [Editor Report · Decision Letter 4]

19 Jul 2024

Dear Mr Fowler, 

On behalf of my colleagues, I am very pleased to inform you that we have agreed to publish your manuscript "Association between surgeon training grade and the risk of revision following unicompartmental knee replacement: an analysis of National Joint Registry Data" (PMEDICINE-D-24-00668R4) in PLOS Medicine.

Prior to publication, could you replace the capital D in the word Data in your full title with a lower case d please? The title should then read: ‘Association between surgeon training grade and the risk of revision following unicompartmental knee replacement: an analysis of National Joint Registry data’. (Apologies, this was my oversight!)

Before your manuscript can be formally accepted, you will also need to complete some formatting changes, which you will receive in a follow up email. Please be aware that it may take several days for you to receive this email; during this time no action is required by you. Once you have received these formatting requests, please note that your manuscript will not be scheduled for publication until you have made the required changes.

PRESS

Sincerely, 

Syba

Syba Sunny, MBBS, MRes, FRCPath 

Associate Editor 

PLOS Medicine

ssunny@plos.org